# Ultrafast in-gel detection by fluorescent super-chelator probes with HisQuick-PAGE

Stefan Brüchert [1,3], Eike F. Joest [1,3], Karl Gatterdam[1,2] & Robert Tampé [1✉]

Polyacrylamide gel electrophoresis (PAGE) and immunoblotting (Western blotting) are the most common methods in life science. In conjunction with these methods, the polyhistidine-tag has proven to be a superb fusion tag for protein purification as well as specific protein detection by immunoblotting, which led to a vast amount of commercially available antibodies. Nevertheless, antibody batch-to-batch variations and nonspecific binding complicate the laborious procedure. The interaction principle applied for His-tagged protein purification by metal-affinity chromatography using N-nitrilotriacetic acid (NTA) was employed to develop small high-affinity lock-and-key molecules coupled to a fluorophore. These multi-valent NTA probes allow specific detection of His-tagged proteins by fluorescence. Here, we report on HisQuick-PAGE as a fast and versatile immunoblot alternative, using such high-affinity fluorescent super-chelator probes. The procedure allows direct, fast, and ultra-sensitive in-gel detection and analysis of soluble proteins as well as intact membrane protein complexes and macromolecular ribonucleoprotein particles.

[1] Institute of Biochemistry, Biocenter, Goethe University Frankfurt, Max-von-Laue-Str. 9, 60438 Frankfurt, Germany. [2] Present address: Institute of Structural Biology, Biomedical Center, University of Bonn, Venusberg Campus 1, 53127 Bonn, Germany. [3] These authors contributed equally: Stefan Brüchert, Eike F. Joest. ✉email: tampe@em.uni-frankfurt.de

Fluorescent probes based on Ni(II)-loaded super-chelator hexavalent *N*-nitriloacetic acid (*hexa*NTA^fluorophore) are versatile, reversible, and nondisturbing tools for fluorescent live-cell labeling of intracellular histidine-tagged proteins and in-situ imaging by super-resolution microscopy (Fig. 1). The super-chelator and a His-tag constitute a small-molecule interaction pair with (sub)nanomolar affinities and high kinetic stability ($k_{off} \approx 10^{-6}$ s$^{-1}$)[1]. Commercially available monovalent *N*-nitriloacetic acid (*mono*NTA) probes lack these properties ($K_D \approx 14$ μM; $k_{off} \approx 1$ s$^{-1}$), and thus cannot be used as antibody-like probes for His-tagged proteins[2,3]. The trivalent *N*-nitriloacetic acid (*tris*NTA) exhibits a nanomolar affinity ($K_D = 9.5 \pm 0.1$ nM) (Fig. 1)[4,5] as well as a low intracellular background in living systems[5–7]. Different dissociation rates for *hexa*NTA and *tris*NTA enable kinetic tracing of fusion proteins with commonly used His-tags[1]. Fluorescent multivalent chelator probes may also offer specific and stable labeling for convenient ultrafast protein detection in polyacrylamide gel electrophoresis (PAGE). Here, we report on various in-gel fluorescence methods summarized as HisQuick-PAGE (Quick His-tag detection). We elaborate that the super-chelator *hexa*NTA^fluorophore allows nearly background-free in-gel detection of His_{10}- or His_{12}-tagged proteins in sodium dodecyl sulfate (SDS)-PAGE. Moreover, we show specific and robust visualization of His_6- to His_{12}-tagged proteins by either *tris*- or *hexa*NTA^fluorophore in native PAGE. Representing a method for the rapid and generic detection of various targets, ranging from soluble proteins to membrane-associated multi-protein complexes and macromolecular assemblies, HisQuick-PAGE offers a versatile and robust alternative to immunoblotting.

## Results

### His-tag labeling by fluorescent NTA probes in SDS-PAGE.

To demonstrate the possibilities of multivalent chelator probes for gel electrophoresis analyses combined with in-gel detection while bypassing any further washing and staining procedures, we used the maltose-binding protein (MBP) harboring a C-terminal His_6-, His_{10}-, or His_{12}-tag as reference. His-tagged proteins were incubated with *tris*NTA^fluorophore or *hexa*NTA^fluorophore (Fig. 2).

We first analyzed specific labeling by discontinuous SDS-PAGE (non-reducing) and in-gel fluorescence utilizing the bright, photostable organic fluorophore Alexa647 for sensitive detection[8]. Immediately after electrophoresis and without any additional washing or staining procedures, we recorded the fluorescence of the labeled proteins. In stark contrast to *tris*NTA^Alexa647, high signal-to-background labeling of His_{10}- and His_{12}-tagged proteins was visible by *hexa*NTA^Alexa647 (Fig. 3a). A slight upshift of the labeled protein was observed by Coomassie staining (Supplementary Fig. 1). To investigate the detection limit of our in-gel labeling method, we titrated various concentrations of the His_{12}-tagged protein preincubated with *hexa*NTA^Alexa647 (450 nM). In non-reducing SDS-PAGE, 10 ng (230 fmol) of MBP-His_{12} were detected by in-gel fluorescence (Fig. 3b, Table 1). In contrast, only 0.3 μg (7 pmol) of protein were visualized by Coomassie staining. We next examined protein labeling under reducing conditions (Fig. 3c). Prior to the addition of *hexa*NTA^Alexa647, His_{12}-tagged proteins were incubated at 95 °C in SDS-loading buffer containing various concentrations of dithiothreitol (DTT). Labeling of His_{12}-tagged proteins was not affected by reducing agent, owing to the high kinetic stability of the super-chelator/His-tag complex. In the presence of 250 mM DTT, which is a concentration commonly used in SDS-PAGE loading buffers, we observed no decrease in the labeling efficiency (Fig. 3c).

### Specific protein labeling in cell lysates by HisQuick-PAGE.

Impelled by the observation that His_6-tagged proteins were not detected by *hexa*NTA^Alexa647 in SDS-PAGE, we rationalized that HisQuick-PAGE (Fig. 2) is less prone to endogenous histidine-rich proteins causing background labeling. Accordingly, we investigated the labeling efficiency and specificity in cell lysates by in-gel fluorescence using *hexa*NTA^Alexa647 (Fig. 4a) and by conventional immunoblotting (Fig. 4b). *E. coli* lysates containing various amounts of His_{12}-tagged proteins were denatured by reducing SDS-PAGE loading buffer at 95 °C, subsequently incubated with *hexa*NTA^Alexa647 at room temperature, and analyzed by SDS-PAGE. Remarkably, *hexa*NTA^Alexa647 labeling of His-tagged proteins in cell lysate under reducing conditions exhibited a similar

**Fig. 1 Chemical structures of the multivalent chelator probes *tris*NTA and *hexa*NTA.** The interaction towards His-tagged proteins is based on an octahedral complex of the NTA moiety and the ligand histidine side chains (X) chelating a Ni(II) ion. Due to six individual complex formations, a single super-chelator *hexa*NTA probe forms kinetically highly stable interactions. As probes for in-gel protein detection during HisQuick-PAGE, the multivalent chelators are covalently coupled to the fluorophore Alexa647 via a carboxy (*tris*NTA) or thiol modification (*hexa*NTA).

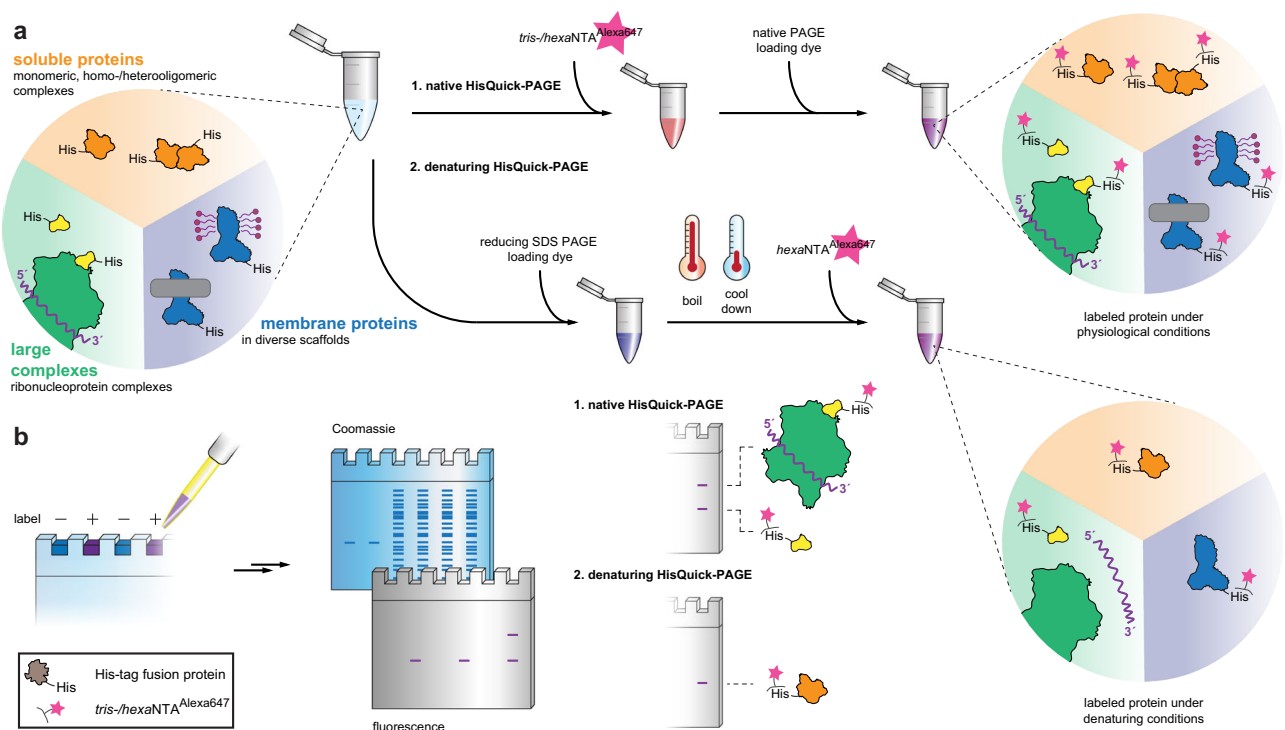

**Fig. 2 HisQuick-PAGE for selective and ultrafast in-gel fluorescence detection. a** Scheme of the labeling procedure under native (1) or denaturing (2) conditions. For native HisQuick-PAGE detection, samples containing $His_{6-12}$-tagged target proteins are labeled either by *tris*NTA[Alexa647] or *hexa*NTA[Alexa647] and immediately detected after electrophoresis. For denaturing HisQuick-PAGE, samples containing $His_{10-12}$-tagged target proteins are incubated with loading dye containing a reducing reagent at 95 °C, labeled by *hexa*NTA[Alexa647], and immediately detected after SDS-PAGE. **b** Quick identification of His-tagged proteins under denaturing conditions or protein-protein interactions, complex formation, or reconstituted membrane proteins under native conditions by in-gel fluorescence. Subsequently, standard Coomassie stains can be applied.

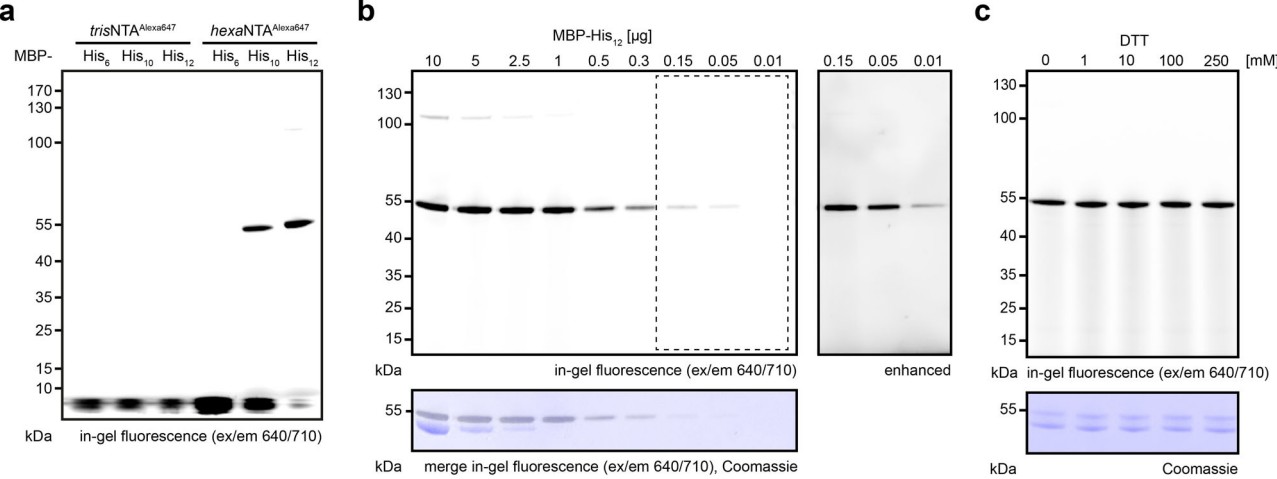

**Fig. 3 Sensitivity of fluorescent chelator probes for HisQuick-PAGE detection. a** Detection of different histidine tags ($His_n$) fused to the maltose-binding protein (MBP) (2 µg, 45 pmol) using *tris*NTA[Alexa647] or *hexa*NTA[Alexa647] (0.45 µM, 7 pmol) by non-reducing SDS-PAGE analysis. **b** Non-reducing SDS-PAGE detection limit of MBP-$His_{12}$ by Coomassie or *hexa*NTA[Alexa647] and shift of labeled protein. **c** HisQuick-PAGE of MBP-$His_{12}$ (2.5 µg, 60 pmol) incubated at 95 °C (10 min) in the presence of increasing concentrations of reducing agent (DTT, 0–250 mM) prior to *hexa*NTA[Alexa647] (0.45 µM, 7 pmol) labeling.

sensitivity compared with isolated proteins. Only high amounts were visible within the lysate by Coomassie staining (Fig. 4a). We did not observe nonspecific *hexa*NTA labeling, demonstrating the high selectivity of HisQuick-PAGE. Hence, this method enables a robust protein detection equivalent to the most common immunoblotting methods (Fig. 2, Table 1)[9,10]. Since background labeling might occur when using different expression hosts, we analyzed bacterial, yeast, insect, and human cell lysates containing 300 ng (7 pmol) of $His_{12}$-tagged proteins by immunoblotting and with an equimolar amount of *hexa*NTA[Alexa647] (7 pmol).

**Table 1 Detection limits of reference protein MBP-His₁₂.**

| Detection system | Detection limit (µg) | Detection limit (pmol) |
|---|---|---|
| InstantBlueTM | 0.3–0.5 | 7–12 |
| Anti-polyHistidine mAb/mouse/Sigma-Aldrich | 0.10–0.15 | 2–3 |
| Anti-6X His tag® antibody (HRP)/rabbit/Abcam | ~0.01 | ~0.2 |
| HisQuick-PAGE | ~0.01 | ~0.2 |

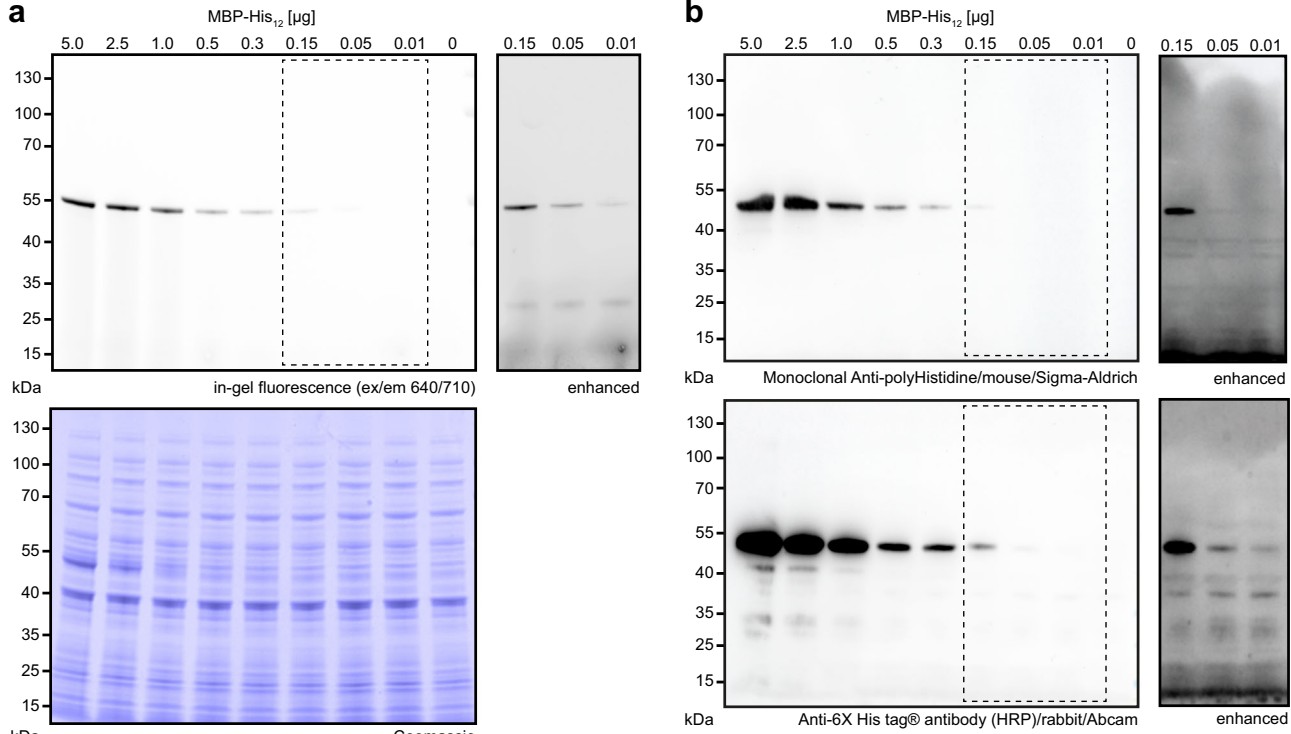

**Fig. 4 Specific and sensitive detection of recombinant proteins in cell lysates by HisQuick-PAGE. a** Different amounts of His-tagged MBP-His₁₂ 10 ng–5 µg (0.23–115 pmol) in *E. coli* lysate (2.5 µg of total protein) revealed the detection limit of *hexa*NTA$^{Alexa647}$ (0.45 µM, 7 pmol). Prolonged exposure visualized 10 ng (230 fmol) of MBP-His₁₂. **b** Detection of MBP-His₁₂ using the same SDS-PAGE conditions by immunoblot and two different anti-His antibodies.

We detected a high signal-to-background labeling, independent of the commonly used expression hosts (Fig. 5a, b).

**Intact protein complexes analyzed by HisQuick-PAGE.** For further implementations, we adapted the HisQuick-PAGE method toward native PAGE and in-gel fluorescence analysis (Fig. 2). We concluded that less harsh conditions enable the detection of recombinant proteins with shorter His-tags and allow labeling by *tris*NTA. The analysis of solubilized or reconstituted membrane proteins in their physiological assembled state is a key feature of the blue native PAGE (BN-PAGE)[11]. However, Coomassie might mask small amounts of diffusely migrating or solubilized proteins. Therefore, we labeled a purified His₁₀-tagged heterodimeric ABC transporter (*Thermus thermophilus* multidrug-resistance proteins A and B, TmrAB) using either *tris*NTA$^{Alexa647}$ or *hexa*NTA$^{Alexa647}$ in a detergent solubilized state (n-dodecyl-β-D-maltoside, DDM) or reconstituted in lipid nanodiscs (membrane scaffold protein MSP1D1)[12,13]. Strikingly, HisQuick-PAGE revealed a background-free labeling of the His₁₀-tagged reference with both the *tris*NTA- and the *hexa*NTA probe. The two detections are not affected by Coomassie, detergent, or lipid nanodiscs (Fig. 5c). Specific His₆-

and His₁₂-tagged reference detection by both fluorescent chelator probes was robustly unaffected in BN-PAGE (Supplementary Fig. 2). As native PAGE allows the visualization of even very fragile complexes[14], we finally combined the sensitive and stable *hexa*NTA$^{fluorophore}$ detection with a clear native PAGE (CN-PAGE) of ribonucleoprotein (RNP) complexes. We utilized the ribosome recycling factor ABCE1-His₆ forming the post-splitting complex with the 30S ribosome. Conditions that trigger the formation of the post-splitting complex 30S·ABCE1 were previously established[15]. ABCE1 and 30S subunits migrated as distinct bands while the formation of the post-splitting complex showed a fluorescent signal at high molecular weight of ~1 MDa (Fig. 5d). Owing to the protein amount used for this HisQuick-PAGE assay, non-complexed ABCE1-His₆ was only visualized by *hexa*NTA$^{Alexa647}$. These results demonstrate that *tris*NTA$^{Alexa647}$ and *hexa*NTA$^{Alexa647}$ label His₆₋₁₂-tagged proteins in native PAGE, in contrast to the denaturing SDS-PAGE displaying a distinct preference for His₁₀- or His₁₂-tags. Moreover, due to a larger range of separation on the native PAGE with around 20 kDa to 2 MDa, a slight upshift caused by the label, as seen for MBP-His₁₂ on the SDS-PAGE, was not observed. Thus, the applicability of the labeling procedure on native PAGE (Fig. 4) proves that

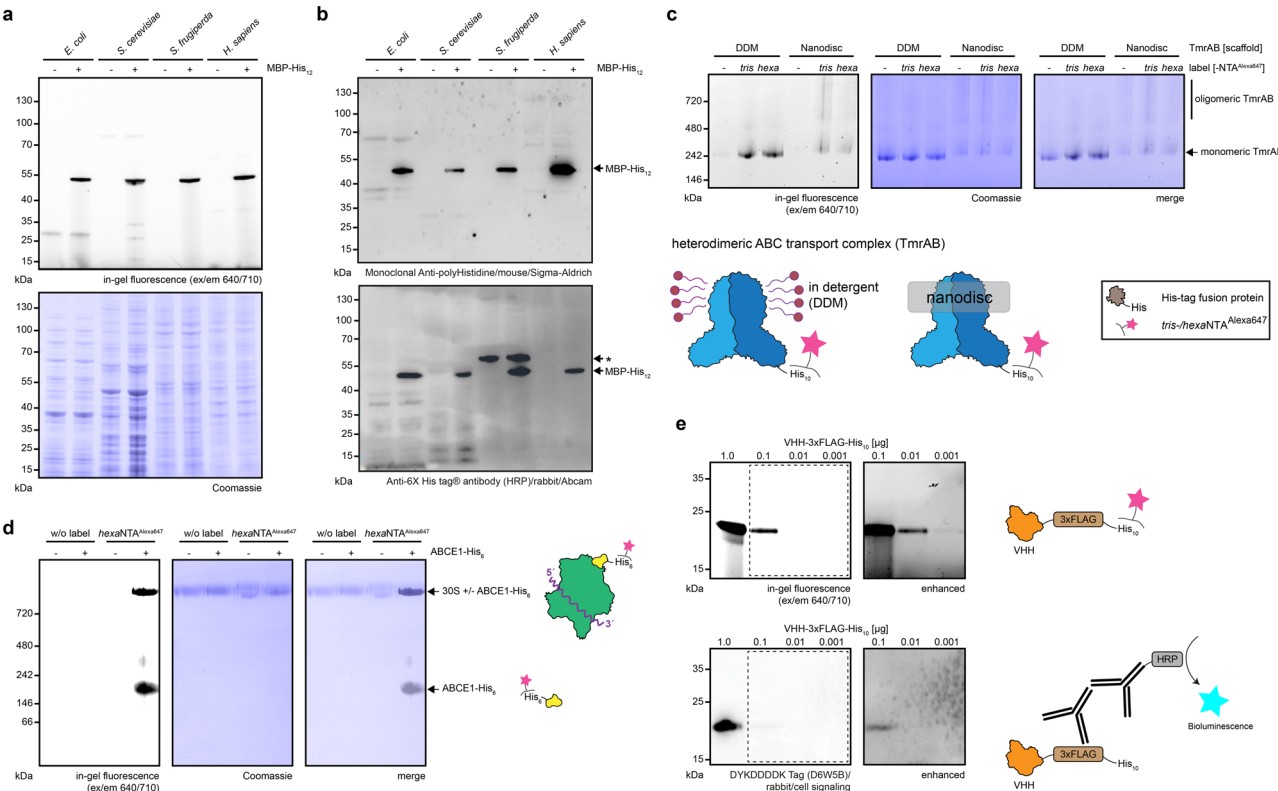

**Fig. 5 Selective protein detection in SDS-PAGE and native PAGE by fluorescent chelator probes. a, b** Detection of His$_{12}$-tagged reference proteins (300 ng, 7 pmol) in cell lysates of bacterial and various eukaryotic origins (2.5 µg total protein) by *hexa*NTA$^{Alexa647}$ (**a**) or immunoblotting using two different primary antibodies (**b**). *, nonspecific band. **c** Detection of a purified His$_{10}$-tagged membrane protein complex (TmrAB, 1.3 µg, 500 nM) in two different commonly used membrane mimetics (DDM and MSP1D1 nanodiscs) were tested on a blue native PAGE by *tris*- and *hexa*NTA$^{Alexa647}$ (450 nM). **d** Visualization of protein complexes containing a His$_6$-tagged component on native PAGE. Non-tagged 30S ribosomal subunit (8 pmol) with fourfold molar equivalence of ABCE1-His$_6$ (32 pmol, 2.3 µg) was used to verify the formation of the 30S·ABCE1-His$_6$ post-splitting complex. Due to the low amount of ABCE1-His$_6$, the His-tagged recycling factor was not visualized by Coomassie staining[15]. **e** Direct contrast of denaturing HisQuick-PAGE using *hexa*NTA$^{Alexa647}$ (450 nM) and 3xFLAG immunoblot detection using various amounts of purified 3xFLAG- and His$_{10}$-tagged reference protein (nanobody C-terminally fused to 3xFLAG- and His$_{10}$-tag).

HisQuick-PAGE is a better alternative than the corresponding, more tedious native PAGE blotting procedure[14,16].

**HisQuick-PAGE versus epitope-based immunodetection.** In addition, we analyzed the detection limit of denaturing HisQuick-PAGE in contrast to an established fusion tag commonly used for immunodetection. For this purpose, we provided a double-tagged anti-GFP nanobody with a C-terminal 3xFLAG-tag followed by a His$_{10}$-tag as a reference protein. Side-by-side, we detected various amounts of reference levels using HisQuick-PAGE that tracks the His$_{10}$-tag or immunoblotting, which targets the 3xFLAG-tag by an appropriate FLAG-epitope antibody. We observed a detection limit of our reference using denaturing HisQuick-PAGE around 10 ng, whereas the detection limit of the anti-FLAG immunoblot was around 100 ng (Fig. 5e). The results show that HisQuick-PAGE is not only comparable with other currently used tags for immunodetection but can also detect even lower amounts of target proteins.

## Discussion

We demonstrated that HisQuick-PAGE enables a direct, fast, and versatile detection for His-tagged proteins and protein complexes. Even the reducing environment by the addition of thiol reagents in SDS-PAGE did not affect the high affinity and kinetic stability of the small interaction pair. Furthermore, we did not detect

background staining caused by nonspecific binding in the cell lysate. Hence, this fast and generic method circumvents blot membrane transfer issues and well-known cross-reactivity problems with His-tag-specific antibodies in immunoblotting. Finally, soluble and membrane proteins either solubilized in detergents or reconstituted in lipid nanodiscs as well as ribonucleoprotein complexes were detected at a high signal-to-background ratio in blue or clear native PAGE by *tris*NTA$^{fluorophore}$ or *hexa*NTA$^{fluorophore}$. Given the dominant use of His-tagged proteins in life science, we anticipate that this set of HisQuick-PAGE applications will extend the range of established methods in the field[17,18].

HisQuick-PAGE requires minimal amounts of *tris*- or *hexa*NTA probes at nanomolar concentrations. Possibly the concentrations may be lowered depending on the protein of interest, especially if His$_{10}$- or His$_{12}$-tagged proteins are detected in native PAGE. However, the optimal design of scaffold, linker, and chelator head is crucial to the high affinity and kinetic stability of the chelator probes[5]. In addition, the exchange of Ni(II) toward Co(III) might further enhance the labeling stability of the His-tag super-chelator complex[19,20]. This study exhibits the useful combination of multivalent NTA probes coupled to the fluorophore Alexa647, while coupling offers multiple customization possibilities, including the unlimited range of organic fluorophores, nanoparticles, or quantum dots. Especially *hexa*NTA$^{fluorophore}$ labeling is unaffected by batch-to-batch variations and more sensitive than commercially available stains, providing a versatile

alternative to immunoblotting. In addition, this very fast and simplified protocol with a minimum number of steps facilitates reproducibility while still allowing a highly specific detection of His-tagged proteins. HisQuick-PAGE will help to expedite the everyday lab load in general, and it will facilitate the rapid expression analysis of recombinant targets and their complexes in particular.

## Methods

**Synthesis of fluorescent multivalent chelator probes**. To label cyclam-Glu-*tris*NTA, the amine functionalized variant of cyclam-Glu-*tris*NTA was coupled to Alexa647 by NHS-labeling, purified by reversed phase C18-HPLC, verified by MALDI-TOF-MS and finally Ni(II) loaded, followed by anion exchange chromatography, according to previous studies[1,5]. The hexavalent NTA was synthesized by Fmoc-based SPPS, coupled to Alexa647 by maleimide-labeling, verified by MALDI-TOF-MS and purified by anion exchange chromatography after Ni(II) loading, as previously elaborated[1].

**Purification of proteins**. MBP-His$_{6-12}$ was expressed in *E. coli* BL21(DE3) and purified via IMAC, as previously described[1]. Clear native PAGE references ABCE1 and ribosomal subunit 30S were acquired using established protocols[15]. The heterodimeric integral membrane protein TmrAB was produced in *E. coli* BL21(DE3), solubilized with DDM, and purified by metal-affinity purification via the C-terminal His$_{10}$-tag fused to TmrA and by size-exclusion chromatography. The MSP1D1 scaffold protein was produced in *E. coli* BL21(DE3) and purified by metal-affinity chromatography via an N-terminal His$_7$-tag. Purified, DDM solubilized TmrAB was reconstituted with bovine brain lipid extract in MSP1D1 nanodiscs, as previously described[13]. The GFP-enhancer nanobody (PDB: 3K1K)[21] decorated with a triple FLAG- and His$_{10}$ tag was kindly provided by Dr. Eric Geertsma.

**Cell lysate preparation**. *Escherichia coli* BL21(DE3), *Spodoptera frugiperda* Sf9, and HeLa Kyoto were pelleted, and resuspended in Dulbecco's phosphate-buffered saline (DPBS, Gibco, pH 7.3) containing phenylmethylsulfonyl fluoride (PMSF) (1 mM) and DNase I (~0.1 mg ml$^{-1}$). Cells were disrupted by sonication, and lysates were cleared by centrifugation (15,000 × g). *Saccharomyces cerevisiae* were pelleted and resuspended in ice-cold lysis buffer (50 mM potassium acetate, 5 mM magnesium acetate, 20 mM HEPES-KOH, pH 7.6) and lysed by vigorous shaking with one cell volume of glass beads for 2 min at 4 °C. Final lysates were obtained by clearing at 4000 × g for 5 min. *Sources of cell lines. Escherichia coli*: One Shot BL21 (DE3), ThermoFisher Scientific, Invitrogen. *Saccharomyces cerevisiae*: Gift by Dr. Peter Koetter (EUROSCARF). *Spodoptera frugiperda*: TriEx Sf9 Cells, Merck, Novagen. *Homo sapiens*: HeLa Kyoto, DSMZ, Leibniz-Institut.

**HisQuick-PAGE labeling**. For non-reducing SDS-PAGE labeling, His$_{6-12}$-tagged proteins were mixed with the multivalent NTA$^{Alexa647}$ probe (450 nM) and incubated for 1 h at room temperature. After labeling, a corresponding amount of SDS-loading buffer was added, the sample was subsequently loaded and electrophoresis performed. For reducing SDS-PAGE labeling, His$_{10-12}$-tagged proteins were mixed with reducing SDS-PAGE buffer containing DTT, denatured at 95 °C for 10 min, cooled down on ice, and incubated with *hexa*NTA$^{Alexa647}$ (450 nM) for 1 h at room temperature. Finally, labeling was detected by in-gel fluorescence. Samples were analyzed using NuPAGE$^{TM}$ 4–12% Bis-Tris protein gels (1.0 mm, 10-well) and the corresponding NuPAGE MOPS SDS running buffer (20×) from Thermo Scientific. The SDS-loading buffer was provided as a fivefold concentrate containing the following components: 0.02% (w/v) bromophenol blue, 30% (v/v) glycerol, 10% (w/v) SDS, 250 mM Tris-HCl, pH 6.8. For reducing conditions, a corresponding amount of DTT was added, in general 250 mM.

**Native HisQuick-PAGE labeling**. His$_{6-12}$-tagged proteins were incubated with *hexa*NTA$^{Alexa647}$ (450 nM) for 1 h at room temperature, before being mixed with native PAGE loading buffer. For clear native PAGE electrophoresis, samples were transferred into glycerol (50%, final). After electrophoresis, labeling was visualized by in-gel fluorescence. Samples were separated using NativePAGE$^{TM}$ 3–12% Bis-Tris protein gels (1.0 mm, 10-well) from Thermo Scientific. *Clear native PAGE buffer components*. Cathode buffer contained 50 mM Tricine, 15 mM Bis-Tris HCl, pH 7.0 at 4 °C. The anode buffer consisted of 50 mM Bis-Tris HCl, pH 7.0 at 4 °C. *Blue native PAGE buffer components*. Cathode buffer contained 50 mM Tricine, 0.004% (w/v) Coomassie G-250 and 15 mM Bis-Tris HCl, pH 7.0 at 4 °C. The anode buffer consisted of 50 mM Bis-Tris HCl, pH 7.0 at 4 °C.

**Gel imaging**. Gels were imaged with a Fusion FX imaging system (Vilber). The Alexa647 signal was detected with an excitation wavelength of 640 nm and a narrow band pass filter at 710 nm to filter the emission fluorescence signal. Coomassie stained gels were imaged by exposure to visible light.

**Immunoblotting**. After running SDS-PAGE, the gel was blotted on a nitrocellulose membrane via semi-dry blotting. The transfer buffer contained 25 mM Tris, 100 mM glycine, 0.1% (w/v) SDS, and 20% (v/v) methanol. The protein was efficiently transferred at a constant voltage of 12 V for 30 min. After transfer, the membrane was blocked for 1 h at room temperature in DPBS (supplemented with Bovine Serum Albumin Fraction V (BSA, 2.5% (w/v)) for Sigma-Aldrich primary antibody (monoclonal anti-polyhistidine antibody produced in mouse) and 5% (w/v) for Abcam primary antibody (Anti-6X His-tag antibody (HRP)). For Cell Signaling Technology primary antibody (DYKDDDDK Tag (D6W5B) rabbit monoclonal antibody), the membrane was blocked for 1 h at room temperature in Tris-buffered saline with TWEEN 20 (TBS-T, pH 7.4) supplemented with skimmed milk powder (5% (w/v)). Blocking was followed by three consecutive washing steps with DPBS buffer (Sigma-Aldrich and Abcam primary antibody) or TBS-T (Cell Signaling Technology primary antibody). Subsequently, the membrane was incubated with primary anti-polyhistidine antibody (Sigma-Aldrich: 1:3000 dilution in DPBS supplemented with 1% (w/v) BSA, Abcam: 1:2000 dilution in DPBS supplemented with 2% (w/v) BSA) or anti-FLAG antibody (Cell Signaling Technology (CST): 1:1000 dilution in TBS-T supplemented with 2% (w/v) skimmed milk powder) over night at 8 °C. Incubation with the primary antibody was followed by three consecutive washing steps with DPBS-T buffer (Sigma-Aldrich and Abcam primary antibody) or TBS-T buffer (Cell Signaling Technology primary antibody). Afterward, the membrane was incubated with a secondary antibody horseradish peroxidase (HRP) conjugate for 1 h at room temperature. Sigma-Aldrich primary antibody was followed by Anti-Mouse IgG (Fc specific)–Peroxidase antibody produced in goat (Sigma-Aldrich): 1:20,000 dilution in DPBS. Abcam primary antibody was followed by anti-Rabbit IgG antibody, (H + L) HRP conjugate produced in goat (Sigma-Aldrich): 1:10,000 dilution in DPBS. Cell Signaling Technology primary antibody was followed by anti-Rabbit IgG antibody, (H + L) HRP conjugate produced in goat (Sigma-Aldrich): 1:10,000 dilution in TBS-T). Subsequently, antibody incubation was followed by three consecutive washing steps with DPBS-T buffer (Sigma-Aldrich and Abcam primary antibody) or TBS-T buffer (Cell Signaling Technology primary antibody) before imaging. In order to detect the chemiluminescence of HRP coupled antibodies, a commercial ECL solution (Clarity Western ECL Substrate, Bio-Rad) was used, and for detection the Fusion FX imaging system (Vilber) was utilized.

**Reporting summary**. Further information on research design is available in the Nature Research Reporting Summary linked to this article.

## Data availability
The authors declare that all data supporting the findings of this study are available from the corresponding author upon request.

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

## Acknowledgements

We thank Holger Heinemann for providing the ABCE1 and ribosome post-splitting complexes, Erich Stefan for providing TmrAB reconstituted in MSP1D1 nanodiscs, Dr. Peter Kötter for providing the yeast lysate, and the entire lab for helpful discussions. We gratefully acknowledge Julian Bruckert and Dr. Jenifer Cuesta-Bernal for the generation and preparation of the dual-tagged nanobody. The German Research Foundation (GRK 1986, SFB 902, and SFB 807 to R.T.) and the Volkswagen Foundation (Az. 91 067 to R.T.) supported this work.

## Author contributions

S.B. and E.J. performed the presented PAGE experiments. S.B. and E.J. produced and purified the detergent solubilized TmrAB sample. Chemical synthesis of *tris*- and *hex-a*NTA^Alexa647 and protein purification of the MBP variants were performed by K.G. The research was supervised by R.T. S.B., E.J., and R.T. analyzed the data and wrote the paper.

## Competing interests

The authors declare no competing interests.
