## [Peer Review File · Communications Biology]

Reviewers' comments:

Reviewer #1 (Remarks to the Author):

Brüchert et al. describe the use of a chelating agent that is linked to Alexa647 for the detection of His-tagged proteins using PAGE. Their system has the advantage over conventional antibody staining in the ease of application and detection. The results are well presented and conclusive and add favorably to the tools developed by Chemical Biologists. The advantage over the use of antibodies is clearly outlined and is a step forward to reduce labor and tedious validation of commercial antibody batches. As such, I support publication in *Communications Biology*.

I suggest one experiment the authors might want to consider to strengthen the manuscript:

In order to demonstrate the advantage over classical western blotting, it would be beneficial to look at a heterologously expressed HA- or FLAG–His-tagged protein (e.g. from mammalian cells) that is being visualized by either and both PAGE and blotting using their system and a commercially available antibody (as performed with Anti-His6-AB in Fig. 1g). Such a side-by-side comparison with one of the most commonly used antibodies might be discussed in terms of epitope size, labor and detection capabilities.

In addition, it is important to the community if the probes are available commercially or upon request.

Johannes Broichhagen
MPIMR

Reviewer #2 (Remarks to the Author):

The authors report the application of a new “superchelator” to detection of modified proteins within PAGE/western blots. This hexaNTA chelator is specific to His10 (or greater) tagged proteins. The authors describe the hexavalent N-nitrilotriacetic acid (hexaNTA) chelator in a previous publication for single cell imaging. This report is different, as the authors use their hexaNTA chelator in more common native/denaturing PAGE gel analysis.

The reported technology is simple, but could greatly reduce gel tearing and error in western blotting, as the technology would make currently needed downstream secondary PAGE gel washing, transfer and staining procedures obsolete. As the hexaNTA chelator is introduced upon loading, the reported technology could significantly change protocol in protein electrophoresis. The result in fig. 1b is particularly intriguing. This result showing hexaNTA zero affinity to the His 6 tag, but high affinity to His 10 tag suggests that one could detect multiple genetically modified proteins of varying his-tag length at the same time, within the same gel (using a future cocktail of different superchelators bearing different dyes).

The detection of His- tagged proteins is normally costly as fluorescent/gold His-tag specific antibodies/PE-secondary antibodies are currently used to directly detect His-tag modified proteins. The authors' hexaNTA small-molecule could replace current systems in His-tag identification. Small molecules are inexpensive, chemically more stable, and are less costly to synthesize vs. antibodies/nanoparticles.

The authors show important controls including non-interference with common PAGE elements including coomassie, DTT and detergents. Figures 1f and 2a are particularly convincing.

Major Corrections:

1) Fig. 1a is confusing. What is the actual structure of trisNTA vs. and hexaNTA? The use of brackets in the figure is non-standard and confusing. The chemical structure of the hexaNTAfluorophore, including the structure of Alexa 647 should be fully drawn out. From figure 1a, it is not clear if the carboxy handle or the thiol handle (or both) are been modified to bear Alexa 647.

2) Page 2. The authors report detection of MBP-His12 at 230 fmol levels by in-gel fluorescence and compare this to Coomassie staining (7000 fmol). Coomassie is not the best control, as coomassie would reveal all proteins, not just His tagged systems. The authors should control their experiment with, OR report detection limits in a table along with, detection limits of contemporary technologies for detecting the his-tag (e.g. <https://www.abcam.com/his-tagged-protein-gold-detection-kit-ab170734.html>). The authors already have detection limit data with Anti-6X His Tag antibody (Fig. 1g).

3) Figures 1e and 2c require more detail in each figure. What are the exact contents of each tube?

Minor Corrections:

-Abbreviations need to be clarified at first mention. Some abbreviations are obvious (DTT) however, for a general audience, all abbreviations should be defined at first mention.

E.g. HexaNTAfluorophore (hexavalent N-nitrilotriacetic acid), DTT, HisQuick-PAGE, SDS-PAGE, MBP

- A reference detailing the synthesis of hexaNTA Alexa 647 is provided in the supporting information. This reference should also be mentioned in the manuscript as well, as details of the chemical synthesis of hexaNTA Alexa 647 are critical for reproducibility.

- (Very minor) An explanation as to what is going on with Fig. 2b Anti-6X His tag antibody on *S. frugiperda* lysate would be appreciated.

Point-to-point reply to the reviewers (COMMSBIO-19-1647-T)

Reviewer #1

Brüchert et al. describe the use of a chelating agent that is linked to Alexa647 for the detection of His-tagged proteins using PAGE. Their systems have the advantage over conventional antibody staining in the ease of application and detection. The results are well presented and conclusive and add favorably to the tools developed by Chemical Biologists. The advantage over the use of antibodies is clearly outlined and is a step forward to reduce labor and tedious validation of commercial antibody batches. As such, I support publication in Communications Biology.

I suggest one experiment the authors might want to consider to strengthen the manuscript: In order to demonstrate the advantage over classical western blotting, it would be beneficial to look at a heterologously expressed HA- or FLAG–His-tagged protein (e.g. from mammalian cells) that is being visualized by either and both PAGE and blotting using their system and a commercially available antibody (as performed with Anti-His6-AB in Fig. 1g). Such a side-by-side comparison with one of the most commonly used antibodies might be discussed in terms of epitope size, labor and detection capabilities.

Reply: We kindly thank the reviewer for the very positive evaluation, the favorable feedback on our work and the helpful minor comments on our study. We extended our manuscript by including a comparison of denaturing HisQuick-PAGE with a commercially available FLAG-epitope antibody on a 3xFLAG-His₁₀-tagged nanobody.

In addition, it is important to the community if the probes are available commercially or upon request.

Reply: Thanks for this helpful comment. We are planning to make the super-chelator compounds commercially available.

Reviewer #2

The authors report the application of a new “super chelator” to detection of modified proteins within PAGE/western blots. This hexaNTA chelator is specific to His₁₀ (or greater) tagged proteins. The authors describe the hexavalent N-nitrilotriacetic acid (hexaNTA) chelator in a previous publication for single cell imaging. This report is different, as the authors use their hexaNTA chelator in more common native/denaturing PAGE gel analysis.

The reported technology is simple, but could greatly reduce gel tearing and error in western blotting, as the technology would make currently needed downstream secondary PAGE gel washing, transfer and staining procedures obsolete. As the hexaNTA chelator is introduced upon loading, the reported technology could significantly change protocol in protein electrophoresis. The result in fig. 1b is particularly intriguing. This result showing hexaNTA zero affinity to the His 6 tag, but high affinity to His 10 tag suggests that one could detect multiple genetically modified proteins of varying his-tag length at the same time, within the same gel (using a future cocktail of different super chelators bearing different dyes).

The detection of His- tagged proteins is normally costly as fluorescent/gold His-tag specific antibodies/ PE-secondary antibodies are currently used to directly detect His-tag modified proteins. The authors’ hexaNTA small-molecule could replace current systems in His-tag identification. Small molecules are inexpensive, chemically more stable, and are less costly to synthesize vs. antibodies/nanoparticles.

The authors show important controls including non-interference with common PAGE elements including Coomassie, DTT and detergents. Figures 1f and 2a are particularly convincing.

Reply: We kindly thank the reviewer for the very approving evaluation and the positive feedback on our work as well as for the helpful minor comments on our study.

Major Corrections:

1) Fig. 1a is confusing. What is the actual structure of trisNTA vs. and hexaNTA? The use of brackets in the figure is non-standard and confusing. The chemical structure of the hexaNTA fluorophore, including the structure of Alexa 647 should be fully drawn out. From figure 1a, it is not clear if the carboxy handle or the thiol handle (or both) are been modified to bear Alexa 647.

Reply: We like to thank the reviewer for this suggestion. The manuscript was originally submitted to Nature Methods as brief communications, limiting the number of figures to two. We now present the chemical structure of trisNTA and hexaNTA in full (see new Figure 1). The coupling groups are integrated in the structure, but the fluorophore is abbreviated as the structure is not officially released. We also highlight the experimental scheme with more clarity (see new Figure 2).

2) Page 2. The authors report detection of MBP-His12 at 230 fmol levels by in-gel fluorescence and compare this to Coomassie staining (7000 fmol). Coomassie is not the best control, as coomassie would reveal all proteins, not just His tagged systems. The authors should control their experiment with, OR report detection limits in a table along with, detection limits of contemporary technologies for detecting the his-tag (e.g. <https://www.abcam.com/his-tagged-protein-gold-detection-kit-ab170734.html>). The authors already have detection limit data with anti-6X His Tag antibody (Fig. 1g).

Reply: We attached a new table with the detection limits. Companies give either no information about a detection limit in Western blots or for different target proteins. Therefore, the numbers shown in the table for the detection limits are the values which we determined using the same target protein.

3) Figures 1e and 2c require more detail in each figure. What are the exact contents of each tube?

Reply: We extended the figure by the addition of another level of detail. The figure now explains in detail how the samples are prepared for both native PAGE as well as SDS PAGE applications. Also, the exact contents of the tubes are depicted besides them where it is important.

Minor Corrections:

1) Abbreviations need to be clarified at first mention. Some abbreviations are obvious (DTT) however, for a general audience, all abbreviations should be defined at first mention. E.g. HexaNTA fluorophore (hexavalent N-nitrilotriacetic acid), DTT, HisQuick-PAGE, SDS-PAGE, MBP

Reply: We changed the abbreviations accordingly.

2) A reference detailing the synthesis of hexaNTA Alexa 647 is provided in the supporting information. This reference should also be mentioned in the manuscript as well, as details of the chemical synthesis of hexaNTA Alexa 647 are critical for reproducibility.

Reply: After formatting the file, the method section is now in the main part and the reference for the synthesis of hexaNTA can now be found more easily.

3) (Very minor) An explanation as to what is going on with Fig. 2b Anti-6X His tag antibody on *S. frugiperda* lysate would be appreciated.

Reply: We assume that an endogenous protein of the *S. frugiperda* lysate leads to cross reactivity with the Abcam anti-His antibody or the corresponding secondary antibody, which is not the case for the Sigma antibody that was used. We do not know about a reference describing which protein causes the reaction but we expect that every different antibody will lead to a unique background pattern.

REVIEWERS' COMMENTS:

Reviewer #1 (Remarks to the Author):

The authors have addressed my raised questions. In addition, they updated the Figures that nicely clarifies the overall manuscript. I therefore support publication in Communications Biology.

Reviewer #2 (Remarks to the Author):

All issues raised on initial review have been addressed.